# Multi-Objective Optimization of an Elastic Rod with Viscous Termination

**Siyuan Xing** [1] and **Jian-Qiao Sun** [2,*]

1   Department of Mechanical Engineering, California Polytechnic State University,
    San Luis Obispo, CA 93047, USA; sixing@calpoly.edu
2   Department of Mechanical Engineering, School of Engineering, University of California Merced,
    Merced, CA 95343, USA
*   Correspondence: jsun3@ucmerced.edu

**Abstract:** In this paper, we study the multi-objective optimization of the viscous boundary condition of an elastic rod using a hybrid method combining a genetic algorithm and simple cell mapping (GA-SCM). The method proceeds with the NSGAII algorithm to seek a rough Pareto set, followed by a local recovery process based on one-step simple cell mapping to complete the branch of the Pareto set. To accelerate computation, the rod response under impulsive loading is calculated with a particular solution method that provides accurate structural responses with less computational effort. The Pareto set and Pareto front of a case study are obtained with the GA-SCM hybrid method. Optimal designs of each objective function are illustrated through numerical simulations.

**Keywords:** multi-objective optimization; genetic algorithm; simple cell mapping; rod vibration; mass–damper–spring termination; impulse response

## 1. Introduction

Structures with viscous boundaries have been applied to diverse areas for vibration reduction [1], sound absorption [2], and boundary control [3]. One recent example is the railway bridge design for high-speed trains where the soil interacting with the bridge has been modeled as mass–damper–spring terminations of the structure [4]. The best design of structures has always been the pursuit of engineers. The optimal structural design must usually accommodate multiple objectives such as the settling time of vibrations, the response amplitude, and the shaping of the frequency response, leading to multi-objective optimization problems (MOPs). This paper presents a study of the multi-objective optimal design of a one-dimensional elastic rod with a mass–damper–spring termination.

The multi-objective nature of the optimization problem leads to a set of optimal solutions called the Pareto set, making set-oriented methods such as simple cell mapping (SCM) [5] suitable for solving such problems. The cell mapping method was initially developed by Hsu [6] for investigating the global behavior of nonlinear dynamical systems, then extended by Sun and his coworkers [7–9] for MOPs. The method seeks optimal solutions by constructing cell mappings based on the local dominance relation of cells in the discretized design space until the optimal solutions are achieved. Although the method is effective for low-dimensional problems, it suffers from the curse of dimensionality for high-dimensional problems because the searching space grows exponentially with the increase of the dimensions.

In terms of solving MOPs with relatively high dimensions, the evolutionary algorithms such as the genetic algorithm (GA) [10], immune algorithm [11], particle swarm optimization (PSO) [12], and ant colony optimization [13] are the mainstream methods for MOPs. The evolutionary algorithms are stochastic methods that mimic the biological evolutionary process using the evolution laws defined based on the Pareto dominance of fitness functions. Such methods can escape the local optima and rapidly discover the

domains containing the solutions. However, the results of evolutionary algorithms can be sensitive to the selection of the hyperparameters.

Recently, Sun and colleagues [5,14] proposed a hybrid method that incorporates NSGAII and simple cell mapping (SCM). The method begins with NSGAII to generate a rough set from several generations such that the domains containing optimal solutions can be outlined. Using the rough set, SCM performs a local recovery method to complete the branches of the Pareto set through iterative refinement of the design space. With the power of NSGAII, the searching domain of the simple cell mapping method has been substantially reduced, making it possible to apply SCM for high-dimensional problems. On the other hand, the SCM method can complement the GA since obtaining outlined optimal domains using the GA is not very sensitive to the selection of the hyperparameters and is much easier than obtaining detailed Pareto optimal solutions using the GA. This can reduce the burden of parameter tuning with the GA. This paper will present a new case study of MOPs by the hybrid GA-SCM method. For more discussions on the advantage of the GA-SCM method and a comparison with different methods, the reader is referred to [5] and the references therein.

To accelerate the MOP algorithms for structural design, a fast and accurate solver that can predict structural response under external loading is needed. Traditional methods such as the finite-element method for calculating structural response can result in considerable computational load. However, obtaining such a solver for structures with viscous terminations is not an easy task. This is because viscous boundary conditions lead to non-self-adjoint boundary value problems that cannot be solved by the traditional method of eigenvalue expansion. To address this issue, several analytical methods have been developed. Hull et al. [15] presented a method that applies modal expansion in the augmented spatial interval where orthogonal eigenmodes exist. Jayachandran and Sun [16] transformed the problem into a self-adjoint boundary value problem in Hilbert space. Oliveto et al. [17] proposed a complex modal expansion method, which requires formulating new orthogonality conditions. Jovannovic [18] formulated the steady-state solution in the form of Fourier series in the state space by reconstructing the differential operator of the equations of motion. Recently, Xing and Sun [19] applied a particular solution method to study the impulsive response of a 1D elastic rod subject to a mass–damper–spring termination.

In this study, we will continue the effort in [19] to optimize the viscous termination of a 1D elastic rod under impulsive loading using the GA-SCM method. The solution of this problem has many potential applications in structural and acoustic design. The dynamic response of the rod will be predicted by the particular solution method. Firstly, we will define the multi-objective optimization problem, followed by the introduction of the GA-SCM hybrid method. Then, we will formulate the impulse response of the structural problem using the particular solution method and introduce the multi-objective functions for the structural optimization problem. We will demonstrate the effectiveness of the GA-SCM method through a case study.

## 2. Multi-Objective Optimization

A continuous multi-objective optimization problem (MOP) can be defined as

$$
\begin{aligned}
\min_{\mathbf{x} \in \mathcal{R}^n} \mathbf{F}(\mathbf{x}), & \\
\text{with } g_i(\mathbf{x}) \leq 0, \ i = 1, \ldots, l, & \\
h_j(\mathbf{x}) = 0, \ j = 1, \ldots, m, &
\end{aligned}
\tag{1}
$$

where $\mathbf{x}$ is a variable of the design space and $g_i$ and $h_j$ are the design constraints. $\mathbf{F}$ is a map comprised of objective functions $f_i$ ($i = 1, 2, \ldots, k$), i.e.,

$$
\mathbf{F}(\mathbf{x}) = \{ f_1(\mathbf{x}), \ldots, f_k(\mathbf{x}) \},
\tag{2}
$$

where $f_i : Q \to \mathcal{R}$. Herein, Q is the feasible set represented by

$$Q = \{\mathbf{x} \in \mathcal{R}^n \mid g_i(\mathbf{x}) \leq 0, \ i = 1, \ldots, l,$$
$$\text{and } h_j(\mathbf{x}) = 0, \ j = 1, \ldots, m\}. \tag{3}$$

The optimal solution of the multi-objective problem is defined in the sense of Pareto optimality, which requires the introduction of the following definitions.

**Definition 1** (Dominance relation [5])**.**

(a) *A vector* $\mathbf{y} \in Q$ *is called strictly dominated (or simply dominated by a vector* $\mathbf{x} \in Q$ *(*$\mathbf{x} \prec \mathbf{y}$*) if*

$$\mathbf{F}(\mathbf{x}) <_p \mathbf{F}(\mathbf{y}) \text{ and } \mathbf{F}(\mathbf{x}) \neq \mathbf{F}(\mathbf{y}),$$

*where* $<_p$ *is an elementwise less-than-or-equal-to relation.*
(b) *A vector* $\mathbf{y} \in Q$ *is called weakly dominated by a vector* $\mathbf{x} \in Q$ *(*$\mathbf{x} \preceq \mathbf{y}$*) if* $\mathbf{F}(\mathbf{x}) \leq_p \mathbf{F}(\mathbf{y})$*.*

The dominance relation defines the "good" solution in the sense of Pareto optimality. This is a strong relation, which can lead to many optimal solutions, because objective functions are considered as equally "good" solutions when they partially satisfy the inequality relations. To define the sets of optimal solutions and their objective functions, we introduce the Pareto set and Pareto front.

**Definition 2** (Pareto point, Pareto set, Pareto front [5])**.**

(a) *A point* $\mathbf{x} \in Q$ *is called Pareto optimal or a Pareto point of* (1) *if there is no* $\mathbf{y} \in Q$ *that dominates* $\mathbf{x}$*.*
(b) *A point* $\mathbf{x} \in Q$ *is called locally (Pareto) optimal or a local Pareto point of* (1) *if there exists a neighborhood* $N_{\mathbf{x}}$ *of* $\mathbf{x}$ *such that there is no* $\mathbf{y} \in Q \cap N_{\mathbf{x}}$ *that dominates* $\mathbf{x}$*.*
(c) *A point* $\mathbf{x} \in Q$ *is called a weak Pareto point or weakly optimal if there exists no* $\mathbf{y} \in Q$ *such that* $\mathbf{F}(\mathbf{y}) <_p \mathbf{F}(\mathbf{x})$*.*
(d) *The set of all Pareto optimal solutions is called the Pareto set, i.e.,*

$$\mathcal{P} = \mathcal{P}_Q := \{\mathbf{x} \in Q \ : \ \mathbf{x} \text{ is a Pareto point of } (1)\}. \tag{4}$$

(e) *The image* $\mathbf{F}(\mathcal{P})$ *of* $\mathcal{P}$ *is called the Pareto front.*

## 3. GA-SCM Hybrid Method

We apply a hybrid method combining genetic algorithms (GAs) and cell mapping methods [14] to solve an MOP with multi-objective performance indices to be defined in Section 4. The hybrid method is initiated with a genetic algorithm (NSGAII) to generate a rough Pareto set in the design space, which is then used by a cell-mapping-based recovery method to seek a complete branch of the Pareto set through iterative refinement of the cellular space of the design parameters, which will be defined in Section 5. The pseudo code of the GA-SCM method is listed in Algorithm 1. The pseudo code for recovering the Pareto optimal solution is listed in Algorithm 2.

As shown in Algorithm 2, the recovery process firstly discretizes the design space and then iterates through elements of the rough Pareto set from the GA or the previous cell partition, performing a one-step simple cell mapping to search local Pareto points. If a cell is mapped to itself (i.e., a local sink is found), then the cell is pushed into the candidate set, followed by an operator to gather nearby solutions into the set to be visited ($S_{\text{tovisit}}$) as long as they dominate some elements in the Pareto set $\mathcal{P}_s$. Otherwise, the destination cell of the cell mapping is pushed to $S_{\text{tovisit}}$. Then, the same iterative procedure will be performed on the set $S_{\text{tovisit}}$ until no new cells can be brought into $S_{\text{tovisit}}$. At last, a dominance check is carried out to remove non-dominant points from the Pareto set. More detail on the method can be seen in [5].

---

**Algorithm 1** GA-SCM algorithm.

---

**Input:** Design space $Q$, cell space partition $N$, refinement partition *sub*, GA population size $n$, objective functions $\mathbf{F}$, refinement number $k$
**Output:** Pareto set $\mathcal{P}_s$, Pareto front $\mathcal{P}_f$
 1: **Initialization** $S_r \leftarrow \text{GA}(n, Q)$ {finding a rough candidate set using the GA}
 2:  $S_c \leftarrow$ cell creation$(S_r, \mathbf{F})$
 3: **while** $i \leq k$ **do** {seeking Pareto set and front using SCM-based local recovery processes}
 4:    $\mathcal{P}_s, \mathcal{P}_f \leftarrow$ recover$(S_c, \mathbf{F}, N, Q)$
 5:    $S_c \leftarrow$ refine$(\mathcal{P}_s, N, sub)$
 6:    $N \leftarrow N \times sub$ {refining cell space}
 7:    $i \leftarrow i + 1$
 8: **end while**

---

**Algorithm 2** SCM-based recovering algorithm.

---

**Input:** Rough Pareto set $\mathcal{P}_s$, rough Pareto front $\mathcal{P}_f$, objective functions $\mathbf{F}$, cell space partition $N$, design space $Q$, max iteration $n$
**Output:** Pareto set $\mathcal{P}_s$, Pareto front $\mathcal{P}_f$ (under the cell space partition $N$)
 1: **Initialization** Discretize design space $Q$ based on the cell space partition $N$
 2:  $S_{\text{visiting}}, S_{\text{visited}} \leftarrow \mathcal{P}_s, S_c \leftarrow \varnothing$ {$S_c$ stores candidate solutions.}
 3: **while** $S_{\text{visiting}} \neq \varnothing$ **do**
 4:    $S_{\text{tovisit}} \leftarrow \varnothing$
 5:    **for** $q \in S_{\text{visiting}}$ **do**
 6:      $C_d \leftarrow$ simple cell mapping$(q, S_{\text{visited}})$
 7:      **if** $C_d \neq q$ **and** $C_d \notin \mathcal{P}_s$ **then**
 8:        $S_{\text{tovisit}} \leftarrow S_{\text{tovisit}} \cup \{C_d\}$
 9:      **else**
10:        $S_c \leftarrow S_c \cup \{C_d\}$
11:        $S_{\text{tovisit}} \leftarrow S_{\text{tovisit}} \cup \{\mathbf{x}|\mathbf{x} \in neighbor(q) \text{ and } \mathbf{x} \prec \mathbf{y} \text{ where } \mathbf{y} \in \mathcal{P}_s\}$ {collecting neighbors that dominate some element(s) in $\mathcal{P}_s$}
12:      **end if**
13:    **end for**
14:    $S_{\text{visiting}} \leftarrow S_{\text{tovisit}}$
15:    $\mathcal{P}_s \leftarrow \mathcal{P}_s \cup S_c, \mathcal{P}_f \leftarrow \mathcal{P}_f \cup \mathbf{F}(S_c)$
16: **end while**
17: $\mathcal{P}_s, \mathcal{P}_f \leftarrow$ dominance check$(\mathcal{P}_s, \mathcal{P}_f)$

---

The detail of the one-step simple cell mapping algorithm is listed in Algorithm 3. The method finds the local optimal solution by checking the dominance relation between a cell and its neighbor. The optimal solution is defined as the most distant cell that dominates the source cell.

---

**Algorithm 3** Simple cell mapping algorithm.

---

**Input:** Objective functions $\mathbf{F}$, cell $C_s$, visited cell set $S_{\text{visited}}$
**Output:** Destination cell $C_d$, visited cell set $S_{\text{visited}}$
 1: $S_{\text{nbr}} \leftarrow neighbor(C_s)$
 2: **for** $N$ in $S_{nbr}$ **do**
 3:    **if** $N \prec C_s$ **and** constraints satisfied **then** {$\mathbf{F}(N)$ can be fetched from visited set directly if $N$ is visited.}
 4:      Store $N$
 5:      $S_{\text{visited}} \leftarrow S_{\text{visited}} \cup \{N\}$
 6:    **end if**
 7: **end for**
 8: $C_d \leftarrow \arg\{\max\|q_s - q_{nbr}\|_2\}$ {$q_s$ and $q_{nbr}$ are the cell centers of $C_s$ and $S_{nbr}$}

---

Given the numerical computation of the impulse response of the rod is the most time-consuming subroutine in this problem, we record all visited cells using a dictionary structure, whose key is the cell index and whose values consist of the multi-objective functions. This way, the algorithm can search for the values in the dictionary with a time complexity $O(1)$, eliminating the repeated computation for cells that have been visited. In addition, the key of a dictionary is unique. Pushing a visited cell to the dictionary will automatically replace the repeated one. Therefore, our implementation, different from that in [14], does not require combining the repeated cells in the visited set.

## 4. Multi-Objective Optimization of Mass–Damper–Spring Termination

### 4.1. Impulse Response

The one-dimensional elastic rod with a mass–damper–spring termination is shown in Figure 1. An impact loading $f(t) = f_0 \delta(t)$ is applied to its free end. Young's modulus, the cross-section area, and the length of the rod are denoted by $E$, $A$, and $L$, respectively. We split total response $u(x, t)$ into the sum of rigid-body and elastic responses such that

$$u(x,t) = u_r(x,t) + u_e(x,t), \text{ with } 0 \le x \le L, t \ge 0. \tag{5}$$

where $u_r$ is the rigid-body response and $u_e$ is the elastic response. From [19], the equations of motion of the system in Figure 1 are in the form

$$\rho A L \ddot{u}_r + M \ddot{u}_r + c \dot{u}_r + k u_r + \tag{6}$$

$$\rho A \int_0^L \frac{\partial^2 u_e(x,t)}{\partial t^2} dx + M \ddot{u}_e(L,t) + c \dot{u}_e(L,t) + k u_e(L,t) = 0,$$

$$c_p^2 \frac{\partial^2 u_e}{\partial x^2} = \ddot{u}_r + \frac{\partial^2 u_e}{\partial t^2}, \tag{7}$$

where $\rho$ is the density and $c_p = \sqrt{E/\rho}$ is the speed of the longitudinal stress wave. The corresponding boundary conditions are

$$EA \frac{\partial u_e(0,t)}{\partial x} = 0, \tag{8}$$

$$EA \frac{\partial u_e}{\partial x}(L,t) = -M[\ddot{u}_r + \frac{\partial^2 u_e}{\partial t^2}(L,t)]$$

$$- c[\dot{u}_r + \frac{\partial u_e}{\partial t}(L,t)] - k[u_r + u_e(L,t)]. \tag{9}$$

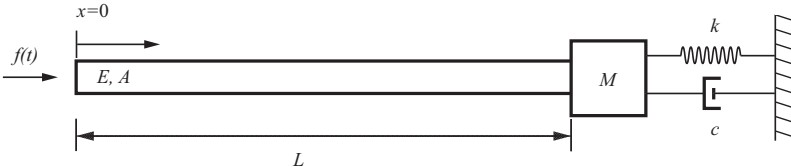

**Figure 1.** A uniform elastic rod with a mass–damper–spring termination. An impact loading $f(t)$ is applied to the free end. The material coordinate system is fixed to the free end of the rod.

The non-homogeneous boundary condition of Equation (9) leads to a non-orthogonal eigenvalue problem. We attack this problem using a method of a particular solution, which expresses the elastic motion $u_e(x,t)$ in the form

$$u_e(x,t) = u_h(x,t) + u_p(x,t), \tag{10}$$

where $u_h(x,t)$ is the homogeneous solution with free–free boundary conditions such that

$$u_h(x,t) = \sum_{i=1}^{n} \phi_i(x)y_i(t),\tag{11}$$

where

$$\int_0^L \phi_i(x)\phi_j(x)dx = \delta_{ij},\tag{12}$$

and $u_p(x,t)$ is the particular solution such that

$$u_p(x,t) = \left(\frac{x}{L}\right)^2 \alpha(t).\tag{13}$$

Substitution of Equations (10)–(13) into Equations (5)–(9) yields a state space form [19]

$$\dot{\mathbf{Z}} = \mathbf{AZ},\tag{14}$$

where

$$\mathbf{Z} = [z(t); \dot{z}(t)],\tag{15}$$

$$\mathbf{A} = \begin{bmatrix} \mathbf{0} & \mathbf{I} \\ -\mathbf{M}^{-1}\mathbf{K} & -\mathbf{M}^{-1}\mathbf{C} \end{bmatrix},\tag{16}$$

$$z(t) = [u_r(t), \alpha(t), y_1(t), y_2(t), \ldots, y_n(t)].\tag{17}$$

The formal solution of Equation (14) reads

$$\mathbf{Z}(t) = e^{\mathbf{A}t}\mathbf{Z}_0,\tag{18}$$

where $\mathbf{Z}_0$ is the initial condition generated from the impulsive input (see Appendix A). The numerical error analysis of the method was performed in [19]. We incorporate this method into the GA-SCM method to optimize the termination of the structure.

*4.2. Objective Functions*

We define the multi-objective performance indices of terminal response as

$$\mathbf{F} = (t_s^{e_3}, |u(L)|_{max}, 1/\delta),\tag{19}$$

where $t_s^{e_3}$ is the settling time of the third elastic mode, $|x(L)|_{max}$ is the maximal absolute displacement at termination, and $\delta$ is the log decrement of the strain response at termination.

$t_s^{e_3}$ is an indirect indicator for the settling time of the rod response. The reason for using $t_s^{e_3}$ is twofold. Firstly, the settling time of higher modes produced by the model cannot properly capture the physical phenomena that the response of high-frequency modes usually decays more rapidly than that of low-frequency modes. Secondly, identifying the settling time of the total response from the numerical simulation could lead to extensive computational load. Therefore, the settling time of the third elastic mode is used and defined in the form

$$t_s^{e_3} = \frac{4}{|Real(\lambda_{e_3})|},\tag{20}$$

where $e_3$ stands for the third elastic mode. The selection of the third mode is based on trial and error.

$\delta$ is also an indirect indicator to estimate the decay of the impact wave. After the impact load is applied, an impulsive wave will be produced at the left terminal and a response wave due to the rigid-body motion will be generated at the right terminal. The two waves will propagate along the rod and be reflected at both ends. Although the strain response is the superposition of two waves, the impact wave dominates the response when it is propagated to the right terminal for the first few times. We define $\delta$ in the form

$$\delta = \frac{1}{n-1} log \frac{|u_x(t_1, L)|}{|u_x(t_n, L)|}, \tag{21}$$

where $t_1$ and $t_n$ represent the first and $n$-th time when the impulse wave is propagated to the right end, respectively. The larger $\delta$ is, the more the impact wave is suppressed. We let $n = 3$ in this study.

## 5. A Case Study

We considered an elastic rod with Young's modulus $E = 10$, density $\rho = 10$, length $L = 2$, cross-section area $A = 0.1$, and excitation force magnitude $f_0 = 1.0$. The design space was chosen as

$$Q = \{\mathbf{x} | \mathbf{x} \in [0.1, 2.0] \times [1.0, 6.0] \times [10, 20]\}, \tag{22}$$

subject to a constraint

$$\delta > 0, \tag{23}$$

where $\mathbf{x}$ is the tuple $(m, c, k)$. We calculated the first 15 s rod response under the impact loading through the numerical integration of Equation (14), because the max displacement appears quickly after impact, and the impact wave dominates the terminal response when it is propagated at the right end during this time period. Thirty elastic modes were adopted, which, based on our observation, are sufficient to approximate the values of performance indices within the design space.

We first discover a rough Pareto set using the NSGAII algorithm with a population size 1000, number of generations 10, and mutation rate 0.05. Other configurations of NSGAII can be seen in Table 1. With the numerical predictor, the NSGAII algorithm was completed in 66 s on a desktop with an Intel core i-7 CPU, producing a rough Pareto set as the input to the SCM method. In the SCM method, the $m - c - k$ design space is discretized into a $10 \times 20 \times 20$ cellular grid as shown in Table 2. The elements of the Pareto set are the cells in the design space. The local search and recovery algorithm are performed twice, the first time with the initial grid and the second time with the refined grid, which divides the initial grid by three. We stop the program after the refinement because the desired resolution $0.06 \times 0.08 \times 0.166$ in the parameter space is achieved. The computational time was 36 s with the initial grid and 2000 s with the refined grid.

**Table 1.** Configuration of NSGAII.

| Encoding | Population | Mutation Rate | Crossover | Generation Number |
|---|---|---|---|---|
| Binary | 1000 | 0.05 | two-point | 10 |

**Table 2.** Configuration of SCM.

| Initial Cell Partition | Sub Partition |
|---|---|
| $10 \times 20 \times 20$ | 3 |

There are 5392 cells in the Pareto set. The Pareto set and front of the mass–damper–spring termination are presented in Figure 2. Generally, either larger stiffness or damping will lead to better design. The majority of optimal design is achieved with either moderate or small mass. The Pareto front can be divided into three regions, labeled in Figure 2b. Region 1 minimizes displacement at the cost of long settling time and moderate damping performance. Region 2 balances the performance of three objective functions. Region 3 achieves premium damping performance at the expense of large displacement and moderate settling time.

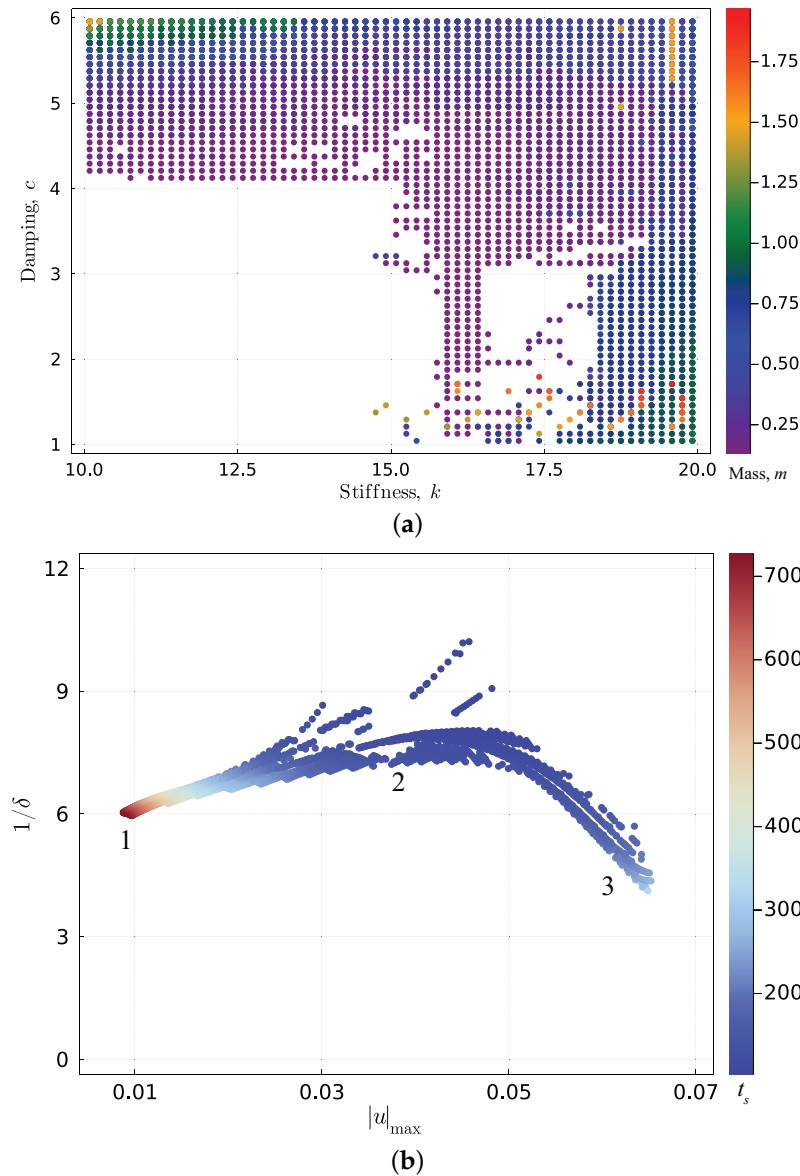

**Figure 2.** The Pareto set and front of the $m - c - k$ termination design of the elastic rod. (**a**) Pareto set. (**b**) Pareto front. Design parameters $m \in [0.1, 2.0]$, $c \in [1.0, 6.0]$, $k \in [10.0, 20.0]$. The labels "1", "2", and "3" indicate the regions where optimal terminal displacement, balanced performance of objective functions, and optimal damping performance are achieved.

The optimal designs of each performance index are presented in Figures 3–5. The corresponding design parameters, as well as performance indices are listed in Table 3.

**Table 3.** Design parameters and performance indices of optimal designs in Figures 3–5.

| Figure | Design Parameters | | | Performance Indices | | |
|--------|--------|--------|---------|--------|----------|-----------------|
|        | $M$    | $c$    | $k$     | $\delta$ | $t_s^{e_3}$ | $\|u(L)\|_{max}$ |
| 3      | 1.4617 | 4.9583 | 18.7500 | 0.1103 | 103.4239 | 0.0482 |
| 4      | 1.0183 | 1.0416 | 19.9167 | 0.2428 | 324.5931 | 0.0648 |
| 5      | 0.1316 | 5.9583 | 19.9167 | 0.0414 | 724.9264 | 0.0088 |

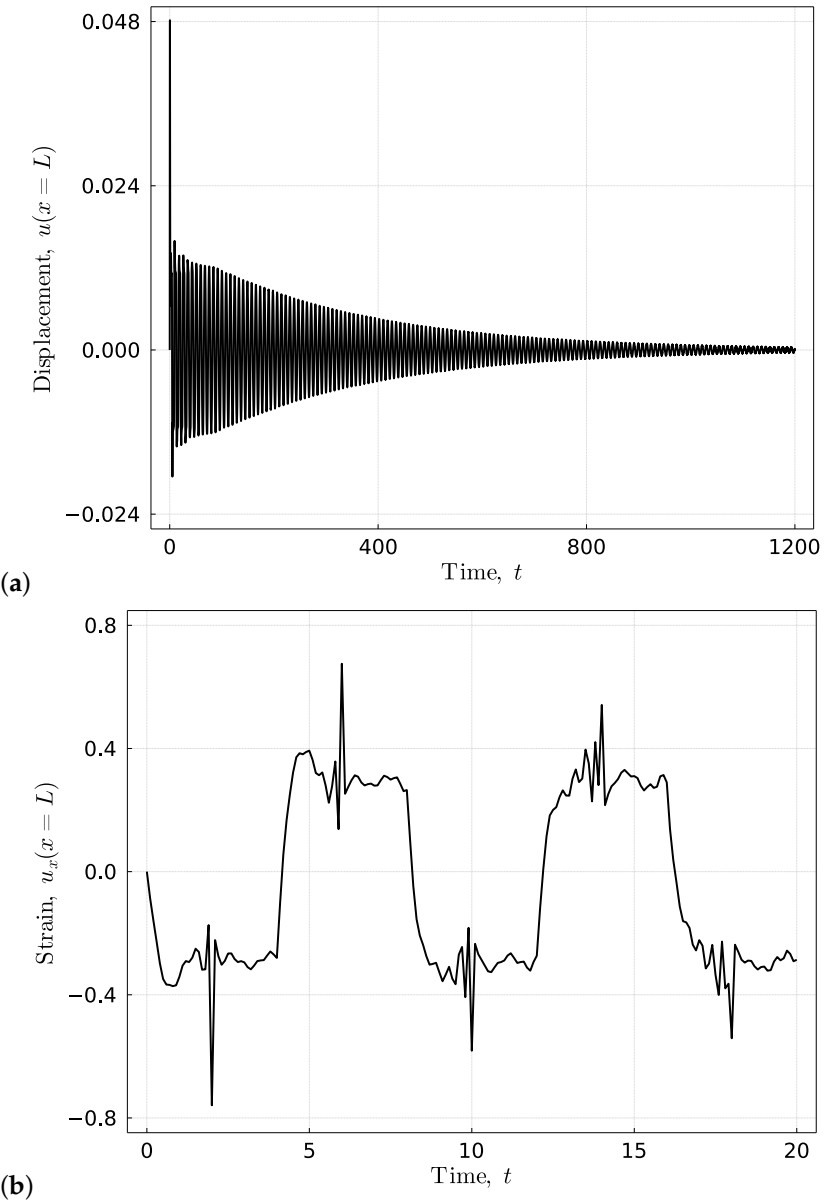

**Figure 3.** The optimal design of settling time. The corresponding (**a**) terminal displacement and (**b**) strain responses of the rod. The response is computed with $N = 30$.

### 5.1. Optimal Design: Minimal Settling Time

Figure 3 shows the optimal design of the settling time. The settling time of the total response approximates 1200 s. While the performance index of the settling time is significantly smaller than this number, it still correctly reflects the trend of the settling time change in comparison to other designs such as those in Figures 4 and 5. The large mass in this design can increase the portion of energy transmitted to the mass after impact, which can be more effectively dissipated through the heavily damped boundary condition.

### 5.2. Optimal Design: Maximal Decay of Impact Wave

The time response of the optimal design maximizing the decay of the impact wave is presented in Figure 4. The impact wave propagates to the right end when $t = 2, 6, 10 \ldots$. The suppression of the impact wave is evident. However, this is at the cost of at least a five-times longer settling time and a slight increase of the maximal displacement. When compared to the other two designs, this design considerably reduces the damping coefficient. This could be attributed to the velocity change of the mounted mass in response to the impact

wave hitting the terminal. Such a change will immediately alter the viscous force produced by the damper, which in turn can lead to higher strain at the terminal. A small damping coefficient can reduce the magnitude of the reflected impact wave.

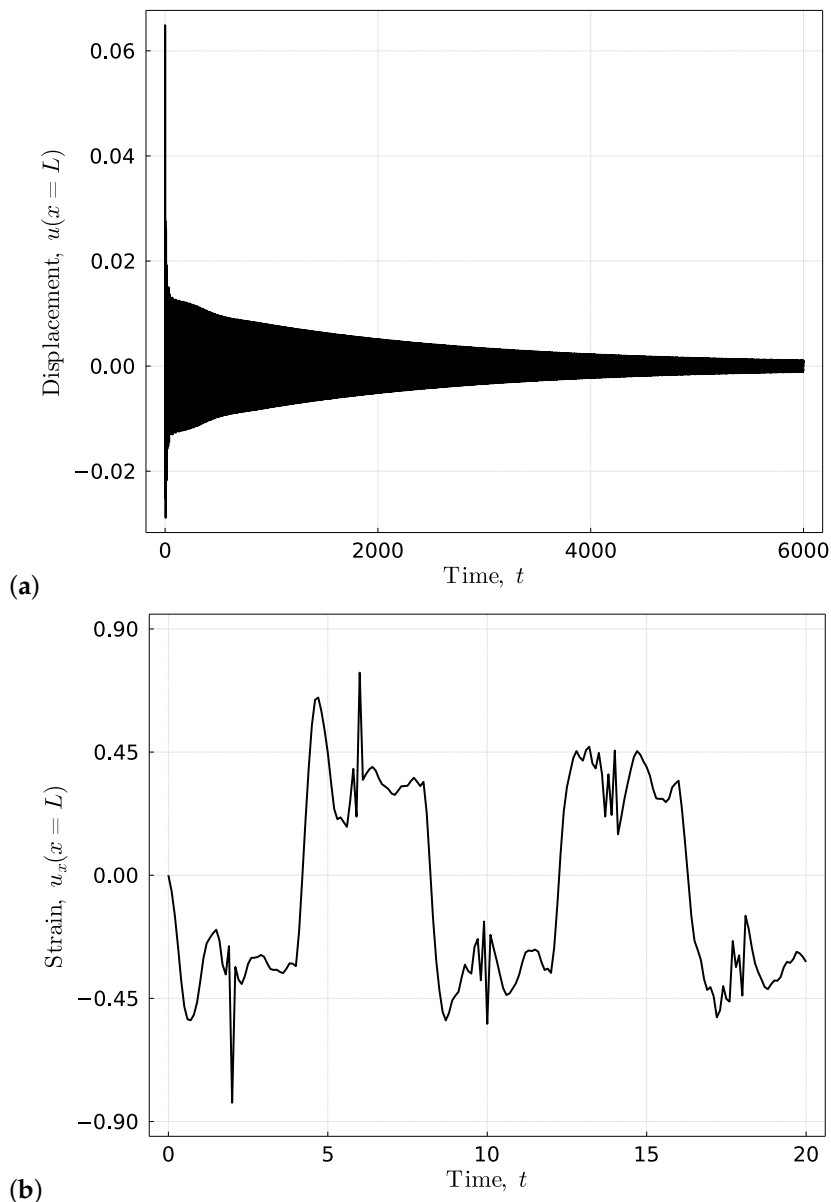

**(a)**

**(b)**

**Figure 4.** The optimal design of the decay of the impact wave. The corresponding (**a**) terminal displacement and (**b**) strain response of the rod. The response is calculated with $N = 30$.

### 5.3. Optimal Design: Minimal Peak Displacement at Termination

The optimal design of terminal peak displacement in Figure 5 has the same stiffness, but much smaller mass and larger damping as the design in Figure 4. This makes sense because the terminal displacement is identical to the displacement of the mounted mass. Using small inertia and large stiffness and damping, one can effectively reduce the maximal terminal displacement. However, smaller inertia also leads to less energy distributed to the mass. Because the energy can only be dissipated through the damper attached to the mass, this choice can also significantly amplify the settling time.

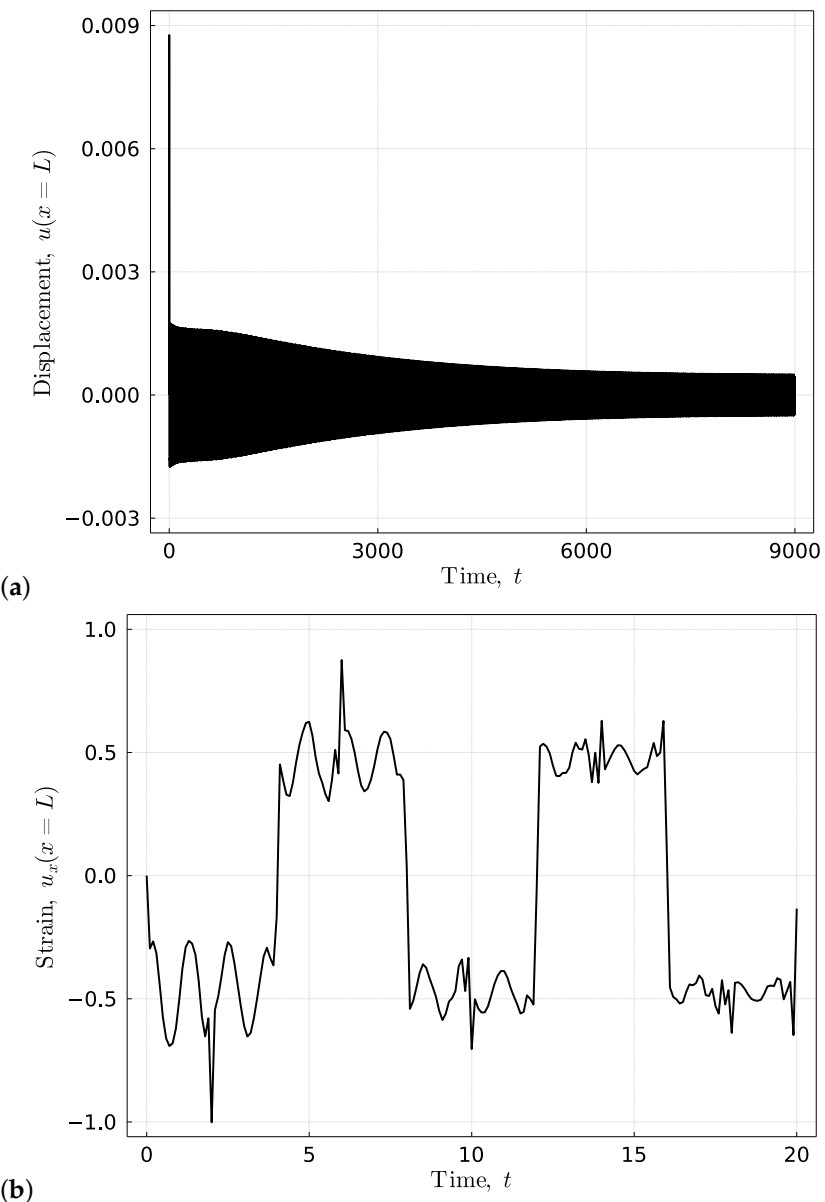

**Figure 5.** The optimal design of the maximal terminal displacement. The corresponding (**a**) terminal displacement and (**b**) strain responses of the rod. The response is calculated with $N = 30$.

## 6. Conclusions

In this paper, a multi-objective optimization problem of the terminal response of an elastic rod with a viscous boundary condition was formulated. The terminal response of the rod was predicted through a computationally effective and accurate particular solution method. The Pareto set and front of the MOP were obtained with the GA-SCM hybrid method. The proposed objective functions can effectively capture the dynamic response of the structure. The optimal design strategies were presented and analyzed. The amount of energy distributed to the terminal mass after impact was significant for the optimization of the terminal design.

The computational load of this work was due to the repeated computations of the impulse response with different parameter sets. Although the solver adopted in this paper can be computationally more effective and accurate than finite-element methods, it still requires a sufficient number of modes to capture the non-smooth impulsive response when highly accurate results are desired. The computational load can be further reduced using a surrogate (metamodel) model [20]. One future direction is to use neural operators such as

DeepONet [21] to approximate the impulsive response, with the neural operator trained using data from the adopted solver.

**Author Contributions:** Conceptualization, methodology, and supervision, J.-Q.S.; software, formal analysis, investigation, and writing—original draft preparation, S.X.; writing—review and editing, J.-Q.S. All authors have read and agreed to the published version of the manuscript.

**Funding:** The first author would like to thank for the release time support from the Donald E. Bently Center for Engineering Innovation at California Polytechnic State University.

**Conflicts of Interest:** The authors declare no conflict of interest.

**Appendix A**

From [19], the initial conditions of Equation (18) are

$$\rho A L \dot{u}_{r0} + M \dot{u}_{r0} = f_0, \tag{A1}$$

$$\dot{u}_{r0} + \sum_{i=1}^{n} \phi_i(0) \dot{y}_{i0} = \frac{f_0}{\rho A}, \tag{A2}$$

$$u_{r0} + \sum_{i=1}^{n} y_{i0} + \left(\frac{x}{L}\right)^m \alpha_0 = 0, 0 \leq x \leq L, \tag{A3}$$

$$\sum_{i=1}^{n} \phi_i(x) \dot{y}_{i0} + \left(\frac{x}{L}\right)^m \dot{\alpha}_0 = 0, 0 < x \leq L, \tag{A4}$$

where $u_{r0} = u_r(0)$, $\alpha_0 = \alpha(0)$, and $y_{i0} = y_i(0)$. Equation (A1) leads to

$$\dot{u}_{r0} = f_0 / (\rho A L + M). \tag{A5}$$

By uniformly sampling spatial points on the rod and applying the least-mean-squares method, the initial conditions of the particular solution and response of elastic modes can be obtained in the form

$$\dot{\mathbf{y}}_0 = (\mathbf{\Phi}^T \mathbf{\Phi})^{-1} \mathbf{\Phi}^T \mathbf{F}, \tag{A6}$$

where $\dot{\mathbf{y}}_0 = [\dot{a}_0, \dot{y}_{10}, \cdots, \dot{y}_{n0}]$, $\mathbf{F} = [f_0 / (\rho A) - \dot{u}_{r0}, 0, 0, ...0]^T$ and

$$\mathbf{\Phi} = \begin{bmatrix} 0 & \phi_1(0) & \phi_2(0) & \cdots & \phi_n(0) \\ (x_1/L)^m & \phi_1(x_1) & \phi_2(x_1) & \cdots & \phi_n(x_1) \\ \vdots & \vdots & \vdots & \ddots & \vdots \\ (L/L)^m & \phi_1(L) & \phi_2(L) & \cdots & \phi_n(L) \end{bmatrix}. \tag{A7}$$

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
