# Peer review of "Multi-Objective Optimization of an Elastic Rod with Viscous Termination"

_mca, doi:10.3390/mca27060094_

Round 1

Reviewer 1 Report

The paper demonstrates the application of a recently established hybrid method for multiobjective optimization – the combination of NSGA-II with simple cell mapping – to a mechanical application: an elastic rod with a spring-damper-fixation at one end. The algorithm and underlying model are described in detail, the solution is shown visually and analyzed in terms of the numerical performance.

Overall, the paper is well written and presents an interesting application. However, since the method is not invented in this article but still rather new, I think it would be very helpful to compare to a more established approach (say, NSGA-II alone), and compare the performance. This might also be very helpful in demonstrating the efficiency. Please find below a list of additional comments:

·         The high computing time is mentioned. Maybe the authors could add a short discussion on surrogate modeling techniques in MO, which have been quite popular lately. These often help mitigating the cost of expensive model evaluations. Examples are

o   “Handling computationally expensive multiobjective optimization problems with evolutionary algorithms : A survey” by Chugh et al.

o   “Derivative-free multiobjective trust region descent method using radial basis function surrogate models”, Berkemeier et al.

·         In Figure 2, what do the numbers 1-3 refer to? I guess the three compromise solutions you study later, but please say so already in the caption. Also, you should visualize the corresponding points in the Pareto set.

·         In table 2: I guess it should be “m”, not “M”.

·         In terms of the comparison in Figs 3-5, I think it would be much more informative if the different compromise solutions were plotted in a single plot, on top of one another. The way it is presented now, a comparison appears to be very difficult.

·         Line 136: Is the grid refinement by a factor of 3 in every dimension?

·         Line 137: How do you define “desired accuracy”?

·         Line 40: “… conditions lead to …”

·         Line 90: “to gather nearby solutions into the set to be visited (Stovisit), as long as they dominate some elements in the Pareto set Ps”

·         Eq. (19): Is it u(L) or x(L)? (Right below, it says x(L))

Line 122: have you defined f(t)? I guess it is a sine, but you just mention f_0

Author Response

Please see the enclosed pdf file.

Reviewer 2 Report

In this work, the authors apply the GA-SCM hybrid method to a viscous boundary condition of an elastic rod. The GA-SCM hybrid method combines cell mapping methods and the well-known NSGA-II. First, the GA is used to find a rough approximation of the Pareto set/front and then the cell mapping methods iteratively refines the solutions.

In general, it is an interesting application of a previously published method to a three-objective optimal control with three decision variables. However, there exist some points to be addressed. In particular, it would be interesting to further motivate the reason for using the GA-SCM hybrid method instead of other methods as well as the contributions of said method. In the following, some comments:

Minor comments:

- In line 64, typically "strictly dominated" and "dominated" stand for different definitions in the literature "<_p" and "<=_p" respectively
- In line 86, "the cellular space" is mentioned please define it

Specific comments:

- Please further motivate the advantages of using the GA-SCM hybrid method for the Elastic Rod with Viscous Termination when compared to other methods from the literature
- In Table 1, please comment on the rationale behind the parameters chosen. The application as posted is a real parameter problem but the encoding is binary. Further, the population size is two orders of magnitude larger than the number of generations. It would be interesting to see what would happen with different configurations.
- In the study case, please mention the parameters used for the SCM part of the algorithm as well as the rationale behind their use
- In the study case, please explicitly mention the number of function evaluations used in each step of the algorithm. Especially for the SCM part.
- In the study case, please compare NSGA-II, SCM, and GA-SCM using the same number of function evaluations. This would help the reader identify the advantages of the selected algorithm

Author Response

Please see the enclosed pdf file.

Reviewer 3 Report

This paper presents a GA-SCM hybrid algorithm with multi-objective optimizations for an elastic rod with viscous termination. It gives a clean and concise analysis of optimal solutions under Pareto optimality and optimal designs of each objective function. The results of the paper are very interesting and will be beneficial to the readers of the journal. For this reason, the reviewer recommends its acceptance for publication.

Author Response

Please see the enclosed pdf file.

Round 2

Reviewer 1 Report

The authors have addressed all my comments. Given their response that there are already several (own) references where the presented method is assessed, I believe that the contribution of this work is really minor without a comparison with other methods. However, I leave it up to the editors whether this is an issue for the journal or not.

Reviewer 2 Report

The authors have addressed my comments and in my opinion the paper can be accepted.